# Soluble IL-2R Levels Predict in-Hospital Mortality in COVID-19 Patients with Respiratory Failure

**DOI:** 10.3390/jcm10184242

**Published:** 2021-09-18

**Authors:** Hye Jin Jang, Ah Young Leem, Kyung Soo Chung, Jin Young Ahn, Ji Ye Jung, Young Ae Kang, Moo Suk Park, Young Sam Kim, Su Hwan Lee

**Affiliations:** 1Division of Pulmonology and Critical Care Medicine, Department of Internal Medicine, Institute of Chest Diseases, Severance Hospital, Yonsei University College of Medicine, Seoul 03722, Korea; rose4359@yuhs.ac (H.J.J.); yimayoung@yuhs.ac (A.Y.L.); chungks@yuhs.ac (K.S.C.); stopyes@yuhs.ac (J.Y.J.); mdkang@yuhs.ac (Y.A.K.); pms70@yuhs.ac (M.S.P.); ysamkim@yuhs.ac (Y.S.K.); 2Division of Infectious Disease, Department of Internal Medicine, Severance Hospital, Yonsei University College of Medicine, Seoul 03722, Korea; comebacktosea@yuhs.ac

**Keywords:** coronavirus disease 2019, soluble interleukin-2 receptor, respiratory failure, mortality

## Abstract

Acute respiratory distress syndrome is the primary cause of death in patients with coronavirus disease 2019 (COVID-19) pneumonia. Our study aims to determine the association between serum markers and mortality in COVID-19 patients with respiratory failure. This retrospective study was conducted in a tertiary care hospital in South Korea. Forty-nine patients with COVID-19, who required high flow nasal cannulation or mechanical ventilation from February 2020 to April 2021, were included. Demographic and laboratory data were analyzed at baseline and on Day 7 of admission. We found that serum creatinine, troponin, procalcitonin, and soluble interleukin-2 receptor (sIL-2R) at baseline were more elevated in the non-survivor group, but were not associated with mechanical ventilator use on Day 7. Older age, PaO_2_/FiO_2_ ratio, lymphocyte and platelet counts, lactate dehydrogenase, IL-6, C-reactive protein, and sIL-2R on Day 7 were significantly associated with mortality. Delta sIL-2R (Day 7–Day 0) per standard deviation was significantly higher in the non-survivor group (adjusted hazard ratio 3.225, 95% confidence interval (CI) 1.151–9.037, *p* = 0.026). Therefore, sIL-2R could predict mortality in COVID-19 patients with respiratory failure. Its sustained elevation suggests a hyper-inflammatory state, and mirrors the severity of COVID-19 in patients with respiratory failure, thereby warranting further attention.

## 1. Introduction

In December 2019, an outbreak of severe acute respiratory syndrome coronavirus 2 (SARS-CoV-2) infection occurred in Wuhan, China. The major cause of death in patients with coronavirus disease 2019 (COVID-19) pneumonia was determined to be multi-organ failure, including respiratory failure, which is a severe disease often requiring invasive ventilation [1,2]. For COVID-19 acute respiratory distress syndrome (ARDS), mortality can range between 26% and 61.5% if the patient is admitted to a critical care setting at any time, and 65.7% in patients who receive mechanical ventilation [2]. 

Previous studies have discovered risk factors associated with poor prognosis in patients with COVID-19, namely, older age [3], D-dimer [3], and procalcitonin [4]. Among the various inflammatory cytokine and chemokine levels assessed in several studies, tumor necrosis factor alpha (TNF-α), interferon-γ-induced protein 10 (IP-10), monocyte chemoattractant protein 1 (MCP-1), chemokine (C-C motif), ligand 3 (CCL-3), and distinct interleukins (IL) (IL-2, IL-6, IL-7, IL-10) have been shown to be significantly associated with disease severity, and in particular, elevated in patients admitted to intensive care units (ICUs) [5]. New cytometric parameters, such as Monocyte Distribution Width (MDW) and Neutrophil-to-Lymphocyte Ratio (NLR), were recently found to correlate well with the clinical severity of COVID-19-associated inflammatory disorder [6,7]. Notably, the serum levels of some interleukins have the potential to discriminate between mild and severe disease, and can possibly be used as prognostic markers [5].

The circulating soluble IL2 receptor (sIL-2R) has been shown to regulate T-lymphocyte activation in various immunological disorders, and increased sIL-2R concentration in plasma predicts a decreased cellular response to IL-2 [8].

The host immune response of COVID-19 suggests an immune signature consisting of elevated serum cytokines (IL-6 and TNF-α); elevated inflammatory serum markers including ferritin, lactate dehydrogenase (LDH), D-dimer, C-reactive protein (CRP), and coagulation factors; impaired interferon responses; and peripheral lymphopenia [3,9,10]. 

In this study, among the several inflammatory markers tested in clinical practice, sIL-2R and IL-6 were investigated for an association with mortality in COVID-19 patients with severe respiratory failure.

## 2. Materials and Methods

### 2.1. Patient Recruitment 

Patient data were retrospectively collected from the ICU of a tertiary hospital in South Korea, from February 2020 to April 2021. We reviewed the medical records of patients who were confirmed to be COVID-19-positive by the detection of viral RNA in their nasopharyngeal secretions or sputum by a real-time reverse transcriptase polymerase chain reaction test. Patients who were diagnosed with respiratory failure and received high flow nasal cannula (HFNC) oxygen therapy or mechanical ventilation were included. Respiratory failure was defined as a PaO_2_/FiO_2_ ratio (the ratio of partial pressure of oxygen in the arterial blood, PaO_2_, to the fraction of inspired oxygen, FiO_2_), of less than 300. 

All patients confirmed to be COVID-19-positive were treated as per the treatment protocol released by the Society of Critical Care Medicine [11]. Patients underwent laboratory tests including CRP, procalcitonin, IL-6, and sIL-2R at the time of admission (Day 0), and on the seventh day of hospitalization (Day 7). The level of sIL-2R was measured using an automated immune chemiluminescent assay with the IMMULITE^®^ 2000 XPi system (Siemens Healthcare GmbH, Erlangen, Germany). To measure the severity of disease, we calculated the Acute Physiology And Chronic Health Evaluation II (APACHE II) score at the time of admission, and a sequential organ failure assessment (SOFA) score at the time of admission and on Day 7. 

### 2.2. Data Collection 

All data were collected from electronic medical records of patients. The following clinical and laboratory data on Day 0 and Day 7 were collected: age; sex; comorbidities; prognosis; P/F ratio; baseline serum markers including CRP, procalcitonin, IL-6, and sIL-2R; SOFA score; and APACHE II score. 

### 2.3. Statistical Analysis

Continuous variables are reported as median with interquartile ranges (IQR, 25th to 75th percentiles), while categorical variables are reported as numbers (percentage). Categorical variables were compared using the chi-square test and continuous variables using either the independent *t*-test or Mann–Whitney U test, according to the non-normal distribution of data.

Bivariate correlations were made by two-tailed Pearson correlation tests and survival curves were shown using Kaplan–Meier analysis. We further investigated the relationship between clinical parameters and mortality using Cox proportional hazard models, with stepwise selection of variables that were found to be significant on univariate regression analysis. In all cases, *p* values < 0.05 were considered statistically significant. All statistical analyses were performed using IBM SPSS Statistics version 25.0 (IBM Corp., Armonk, NY, USA).

### 2.4. Ethics

This research protocol was approved by the Institutional Review Board of Severance Hospital, South Korea (IRB No. 4-2021-0305). The study design was approved by the appropriate ethics review board, and the requirement to obtain informed patient consent was waived due to the retrospective nature of the study. 

## 3. Results

Forty-nine patients with respiratory failure, aged ≥ 18 years, who required HFNC or mechanical ventilator during the study period, were included in this study (Figure 1). 

Table 1 shows the baseline characteristics of patients at the time of ICU admission. The median age of patients was 71.0 years and proportion of males was 59.2%. There was no significant difference in the age, sex, presence of hypertension, IL-6, and the PaO_2_/FiO_2_ ratio at the time of ICU admission between the survivor and non-survivor groups. However, the proportion of diabetes mellitus was significantly higher in the non-survivor group than in the survivor group (60% vs. 25.6%, *p* = 0.041). Furthermore, sIL-2R, creatinine, troponin, and procalcitonin were significantly higher in the non-survivor group than in the survivor group. The median sIL-2R was 1159.0 U/mL (IQR 1099.5–1376.5) in the non-survivor group and 853.0 U/mL (IQR 727.5–1150.5) in the survivor group (*p* = 0.010). 

On the seventh day of hospitalization (Table 2), there were statistically significantly differences in the lymphocyte counts, platelet counts, creatinine, LDH, procalcitonin, CRP, IL-6, and sIL-2R between the two groups. Notably, in the non-survivor group, the sIL-2R on Day 7 was 1313.0 U/mL (IQR 1001.3–1963.5), which was higher than that on Day 0, which was 1159.0 U/mL (IQR 1099.5–1376.5); however, in the survivor group, sIL-2R levels were decreased on Day 7 (845.0 U/mL, IQR 563.8–1061.3, *p* = 0.001) compared with that on Day 0 (853.0 U/mL, IQR 727.5–1150.5, *p* = 0.010). IL-6 concentrations decreased during the course of hospitalization in both groups (survivor group: 17.9 pg/mL, IQR 6.1–51.9; non-survivor group: 100.7 pg/mL, IQR 41.1–244.6, *p* = 0.001). 

Figure 2 shows the scatter plot of PaO_2_/FiO_2_ ratio, IL-6, and sIL-2R on Days 0 and 7. On Day 0, there was no significant correlation between IL-6 and PaO_2_/FiO_2_ ratio (*p* = 0.738, Figure 2a), as well as sIL-2R and PaO_2_/FiO_2_ ratio (*p* = 0.340, Figure 2b). On Day 7, however, sIL-2R was significantly correlated with P/F ratio (*p* = 0.013, Figure 2d); however, IL-6 was not significantly correlated (*p* = 0.167, Figure 2c).

Table 3 shows the comparison of baseline characteristics between the invasive mechanical ventilation (MV) and non-MV groups. IL-6 and sIL-2R were not associated with MV (*p* = 0.855 and *p* = 0.087, respectively). 

Table 4 shows the association between variables including serum markers and mortality, calculated using Cox regression analysis. The difference in platelet counts between Days 0 and 7 was related to high mortality rates (hazard ratio, HR 0.994, *p* = 0.019), and the HR per standard deviation (SD) of ΔsIL-2R was 3.145 (*p* = 0.022) on univariate analysis. The higher the ΔIL-6 per SD, the higher was the mortality rate, though it was not statistically significant (HR 1.664, *p* = 0.107). Even after adjusting for several variables using multivariate analysis, ΔsIL-2R was associated with higher mortality, which adjusted the HR to 3.225 (95% confidence interval, CI: 1.151–9.037, *p* = 0.026).

Figure 3 shows the Kaplan–Meier survival curve of IL-6 and sIL-2R. The cut-off value of IL-6 obtained from the receiver operating characteristic curve was 191 pg/mL (Area Under the Curve, AUC: 0.687, sensitivity: 50.0, specificity: 87.2), and it was significantly related to a higher mortality rate (*p* = 0.006, Figure 3a). The patient group with sIL-2R above 1083 U/mL (AUC: 0.718, sensitivity: 80.0, specificity 71.8) was significantly associated with mortality (*p* = 0.040, Figure 3b).

## 4. Discussion

In the present study, we evaluated the relationship between laboratory variables associated with immune response and the clinical features observed in patients with severe COVID-19. Our analysis demonstrated that several immunological parameters are associated with severity and mortality. Among them, IL-6 and sIL-2R were found to be clinically significant in COVID-19 patients with respiratory failure. IL-6 was not associated with mortality at admission, but sIL-2R was associated. Both were associated with mortality on the seventh day of admission, and sIL-2R was also correlated with the PaO_2_/FiO_2_ ratio, which reflects the severity of COVID-19 pneumonia. However, sIL-2R was not associated with the use of mechanical ventilation. 

Previous studies have demonstrated the relationship between laboratory markers and clinical outcomes in patients with COVID-19. IL-6, for instance, has been analyzed as a prognostic marker [12,13,14]. Thrombocytopenia is also known as a factor associated with the risk of severe disease and mortality in patients with COVID-19, and thus serves as a clinical indicator of worsening illness during hospitalization [15,16]. Recent studies have suggested an association of sIL-2R and disease severity [9]. Hou et al. [17] found higher IL-2R-to-lymphocyte ratio to be related to the severity of COVID-19; however, the level of IL-2R itself was not statistically significant for the prediction of severe COVID-19. Our study results confirmed this finding and showed that the level of sIL-2R is associated with survival, as well as disease severity and that this was more applicable and easier to use in clinics, indicated by the PaO_2_/FiO_2_ ratio in patients with COVID-19. However, it did not show any association with mechanical ventilator use.

Much of the mortality has been associated with “cytokine storm syndrome” in patients admitted to the hospital with COVID-19 pneumonia [18]. Defining the COVID-19 cytokine storm syndrome has been challenging, but early reports propose combinations of clinical (e.g., fever) and laboratory (e.g., hyperferritinemia) features in determining patients most likely to benefit from treatments for the cytokine storm syndrome [19,20]. These indicators mirror the level of systemic hyper-inflammation, lung inflammation, and the severity of organ damage [21]. One of the first strategies toward treating the COVID-19 cytokine storm syndrome was targeting IL-6 [22]. A recombinant humanized anti-IL-6 receptor monoclonal antibody (Tocilizumab) is being used for treating COVID-19 patients that inhibits the binding of IL-6 to both membrane and soluble IL-6 receptors, blocking IL-6 signaling and reducing inflammation [23]. A recent randomized controlled study demonstrated the efficacy of tocilizumab in improving survival and clinical outcomes [23]. Our study did not show a significant relationship between the initial level of IL-6 and survival outcome in COVID-19 patients. This may have been due to the small sample size; however, the level of IL-6 on the seventh day of admission was associated with mortality. It is suggested that if the level of IL-6 is consistently high, clinical outcomes could be worse; therefore, close observation is needed from the time of initial diagnosis of COVID-19.

Another notable marker among the many inflammatory markers is sIL-2R. SARS-CoV-2 has been known to lead to hyperinflammation as well as T-cell deficiencies, associated with life-threatening organ dysfunction [24,25]. Circulating sIL-2R is actively involved in the regulation of T-cell immune responses and is, therefore, suggested to have a role in disease expression [26]. Serum levels of sIL-2R are significantly higher in patients with Kawasaki disease [27], who suffer from a systemic inflammatory disease closely associated with infections [28], and autoimmune diseases [29]. A recent study demonstrated that the concentration of sIL-2R in the blood may be a co-indicator of the severity of COVID-19 with lymphopenia, through IL-2 signaling inhibition [30]. The cut-off value of sIL-2R related to mortality was 1083 U/mL in our study, which was similar to that reported by another recent study, i.e., 1060 U/mL [31]. Taken together, our study further supports the presence of elevated serum sIL-2R in COVID-19 patients with respiratory failure and its correlation with poor prognosis.

We recognize that this study did have several limitations: in particular, the small sample size and the use of retrospective data. However, all registered patients were treated with the same protocol, and data on inflammatory markers such as IL-6 and sIL-2R were uniformly collected. Second, since the subjects of this cohort study were those with respiratory failure, it was not possible to confirm whether this result could be applied to patients with mild disease. Moreover, since these serum markers were not related to the application of mechanical ventilation, they did not help determine the timing of initiation of mechanical ventilation in a clinical setting.

Irrespective of these limitations, we determined that circulating sIL-2R was associated with mortality and disease severity as indicated by the PaO_2_/FiO_2_ ratio. This knowledge can shed light on the cellular mechanism and immune regulation involved in the pathogenesis of COVID-19; furthermore, it can give impetus for discovery of treatment options in the future. A large cohort study is needed to validate our findings.

## 5. Conclusions

In summary, the most relevant result of this study was that sIL-2R was significantly associated with mortality and disease severity in COVID-19 patients with respiratory failure, even after adjusting for several variables. Considering this association between the serum level of sIL-2R and the clinical outcome in COVID-19 patients with respiratory failure, sequential surveillance of sIL-2R and close observation would be needed.

## Figures and Tables

**Figure 1 jcm-10-04242-f001:**
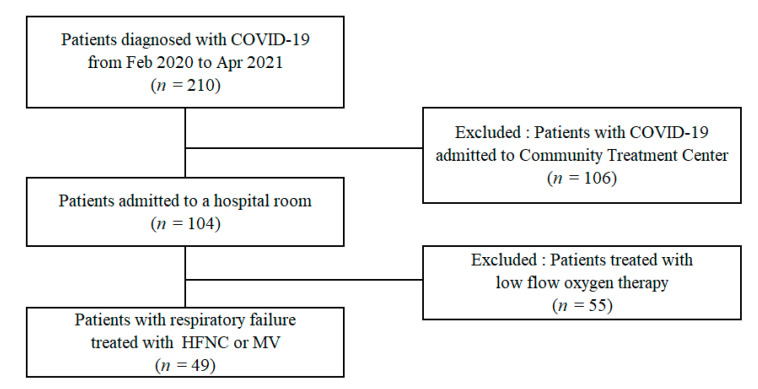
Patient recruitment flowchart. HFNC: high flow nasal cannula; MV: mechanical ventilation.

**Figure 2 jcm-10-04242-f002:**
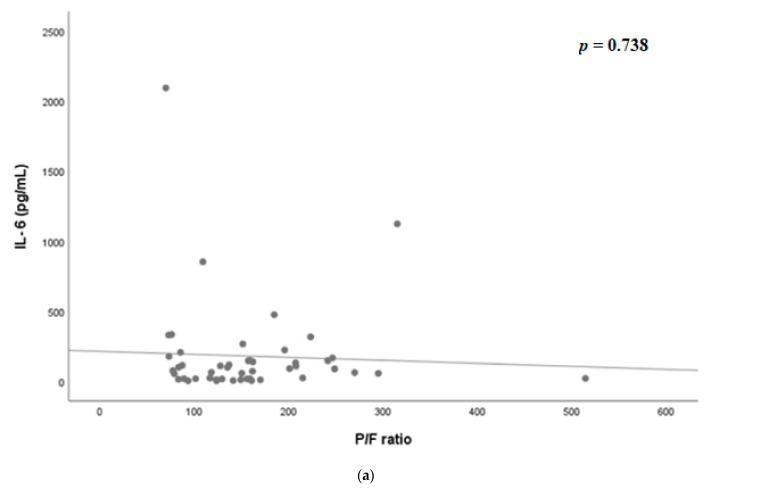
(**a**) Scatter plot of PaO_2_/FiO_2_ ratio and IL-6 on the day of admission. (**b**) Scatter plot of PaO_2_/FiO_2_ ratio and soluble interleukin-2 receptor on the day of admission. (**c**) Scatter plot of PaO_2_/FiO_2_ ratio and IL-6 on hospital day 7. (**d**) Scatter plot of PaO_2_/FiO_2_ ratio and soluble interleukin-2 receptor on hospital day 7. IL-6, interleukin-6; IL2 R, interleukin-2 receptor.

**Figure 3 jcm-10-04242-f003:**
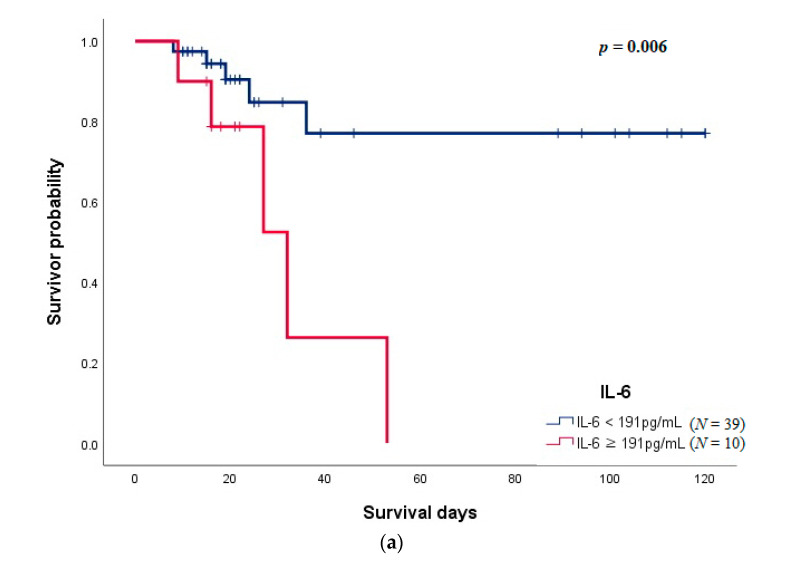
(**a**) Kaplan–Meier survival curves according to IL-6 on day 1 of admission. (**b**) Kaplan–Meier survival curves according to soluble interleukin-2 receptor on day 1 of admission.

**Table 1 jcm-10-04242-t001:** Baseline characteristics of patients on the first day of admission.

Variable	Survivors (*n* = 39)	Non-Survivors (*n* = 10)	Total (*n* = 49)	*p*-Value
Age (years)	70.0 (64.0, 74.0)	78.0 (65.0, 84.8)	71.0 (64.5, 77.0)	0.063
Sex, male (%)	22 (56.4)	7 (70.0)	29 (59.2)	0.517
BMI, kg/m^2^	24.0 (21.4, 25.7)	20.4 (18.8, 25.1)	24.0 (20.5, 25.5)	0.132
Hypertension	22 (56.4)	8 (80.0)	30 (61.2)	0.176
Diabetes	10 (25.6)	6 (60.0)	16 (32.7)	0.041
PaO_2_/FiO_2_, kPa	149.8 (100.9, 184.3)	125.1 (86.3, 229.2)	148.7 (96.9, 197.9)	0.951
ARDS				
Mild	9 (23.1)	3 (30.0)	12 (24.5)	
Moderate	21 (53.8)	4 (40.0)	25 (51.0)	
Severe	9 (23.1)	3 (30.0)	12 (24.5)	
WBC (10^3^/uL)	8.6 (6.7, 12.6)	6.2 (4.5, 8.5)	8.1 (6.0, 11.0)	0.080
Lymphocyte (10^3^/uL)	0.8 (0.5, 1.0)	0.6 (0.4, 1.0)	0.7 (0.5, 1.0)	0.517
Platelet (10^3^/uL)	185.0 (136.0, 246.0)	127.5 (109.5, 170.8)	181.0 (126.0, 232.0)	0.022
Albumin (g/dL)	3.7 (3.3, 3.9)	3.6 (3.1, 3.8)	3.6 (3.3, 3.9)	0.358
Total bilirubin (mg/dL)	0.7 (0.5, 0.9)	0.7 (0.5, 1.3)	0.7 (0.5, 1.0)	0.566
Bicarbonate (mmol/L)	23.0 (21.0, 25.0)	22.0 (19.0, 23.5)	23.0 (21.0, 24.8)	0.164
Creatinine (mg/dL)	0.7 (0.6, 1.1)	1.0 (0.9, 2.3)	0.9 (0.6, 1.1)	0.012
Troponin (pg/mL)	12.0 (7.0, 17.3)	33.0 (17.5, 75.8)	12.0 (8.0, 25.3)	0.001
LDH (IU/L)	527.0 (427.0, 639.0)	494.0 (379.8, 652.3)	514.0 (420.0, 641.5)	0.779
Procalcitonin (ng/mL)	0.3 (0.1, 0.5)	0.5 (0.4, 1.2)	0.3 (0.1, 0.5)	0.009
Fibrinogen (mg/dL)	471.0 (390.8, 613.5)	433.0 (305.5, 521.5)	458.0 (381.0, 599.0)	0.213
D-dimer (ng/mL)	480.0 (273.0, 1386.0)	599.5 (328.5, 3858.5)	494.0 (294.0, 1288.5)	0.333
Ferritin (ng/mL)	787.8 (443.6, 1485.7)	1056.0 (422.1, 1688.0)	801.3 (446.1, 1485.6)	0.505
CRP (mg/L)	107.6 (59.7, 155.9)	140.3 (55.7, 218.4)	112.2 (59.7, 180.3)	0.371
IL-6 (pg/mL)	65.1 (17.8, 147.1)	160.1 (81.4, 320.7)	89.3 (19.4, 158.4)	0.072
Soluble IL2 receptor (U/mL)	853.0 (727.5, 1150.5)	1159.0 (1099.5, 1376.5)	905.5 (740.3, 1230.8)	0.010
Invasive ventilation	11 (28.2)	4 (40.0)	14 (32.6)	0.582
SOFA score	3.0 (0.3, 8.3)	3.0 (2.8, 4.0)	3.0 (3.0, 3.0)	0.385
APACHE II score	24.0 (20.0, 26.0)	24.5 (19.0, 28.5)	24.0 (20.0, 26.5)	0.391
Hospital day	18.5 (15.0, 24.3)	21.5 (13.5, 33.0)	19.0 (15.0, 25.3)	0.695

Abbreviations: BMI, body mass index; ARDS, acute respiratory distress syndrome; WBC, white blood cell; LDH, lactate dehydrogenase; CRP, C-reactive protein; IL-6, interleukin-6; IL-2, interleukin-2; SOFA, Sequential Organ Failure Assessment score; APACHE II, Acute Physiology And Chronic Health Evaluation II.

**Table 2 jcm-10-04242-t002:** Characteristics of patients on the seventh day of admission.

Variable	Survivors (*n* = 39)	Non-Survivors (*n* = 10)	Total (*n* = 49)	*p*-Value
PaO_2_/FiO_2_, kPa	216.0 (165.4, 290.1)	140.6 (71.3, 160.5)	206.7 (141.7, 264.5)	0.001
ARDS				
Mild	22 (56.4)	2 (20.0)	24 (49.0)	
Moderate	15 (38.5)	5 (50.0)	20 (40.8)	
Severe	0 (0.0)	3 (30.0)	3 (6.1)	
WBC (10^3^/uL)	11.7 (7.0, 13.2)	16.2 (7.7, 21.1)	11.8 (7.4, 16.0)	0.189
Lymphocyte (10^3^/uL)	0.8 (0.5, 1.2)	0.4 (0.2, 0.5)	0.7 (0.4, 1.2)	0.005
Platelet (10^3^/uL)	272.0 (187.0, 362.0)	117.0 (72.0, 173.8)	242.0 (152.0, 353.0)	0.001
Albumin (g/dL)	2.9 (2.7, 3.2)	2.8 (2.7, 3.0)	2.9 (2.7, 3.2)	0.235
Total bilirubin (mg/dL)	0.6 (0.4, 0.9)	0.7 (0.4, 1.0)	0.6 (0.4, 0.9)	0.795
Bicarbonate (mmol/L)	25.0 (24.0, 29.0)	24.0 (23.0, 28.8)	25.0 (23.0, 29.0)	0.416
Creatinine (mg/dL)	0.6 (0.5, 0.7)	0.9 (0.7, 2.0)	0.6 (0.5, 0.8)	0.001
LDH (IU/L)	402.0 (320.5, 475.3)	532.0 (432.5, 637.5)	420.0 (334.0, 532.0)	0.013
Procalcitonin (ng/mL)	0.1 (0.0, 0.1)	0.2 (0.1, 0.7)	0.1 (0.1, 0.2)	0.008
Fibrinogen (mg/dL)	384.5 (348.5, 434.3)	440.5 (256.8, 543.0)	399.0 (349.0, 458.0)	0.634
D-dimer (ng/mL)	982.0 (360.0, 3591.0)	1877.5 (454.8, 6152.0)	987.5 (373.8, 3787.0)	0.338
Ferritin (ng/mL)	548.1 (334.0, 846.5)	592.9 (410.0, 770.5)	571.1 (346.1, 840.1)	0.746
CRP (mg/L)	25.6 (8.7, 52.3)	65.5 (35.5, 112.0)	35.3 (10.1, 54.4)	0.016
IL-6 (pg/mL)	17.9 (6.1, 51.9)	100.7 (41.1, 244.6)	21.6 (8.0, 87.5)	0.001
Soluble IL2 receptor (U/mL)	845.0 (563.8, 1061.3)	1313.0 (1001.3, 1963.5)	949.5 (640.0, 1126.3)	0.001
Invasive ventilation	22 (56.4)	7 (70.0)	27 (59.2)	0.617

Abbreviations: ARDS, acute respiratory distress syndrome; WBC, white blood cell; LDH, lactate dehydrogenase; CRP, C-reactive protein; IL-6, interleukin-6; soluble IL-2 receptor, soluble interleukin-2 receptor.

**Table 3 jcm-10-04242-t003:** Patient demographics according to the application of invasive ventilation.

Variable	Non-MV Group (*n* = 20)	MV Group (*n* = 29)	*p*-Value
Age (years)	71.0 (65.3, 78.6)	70.0 (63.0, 75.5)	0.653
Sex, male (%)	10 (50.0)	19 (65.5)	0.377
BMI, kg/m^2^	24.2 (20.3, 25.9)	23.8 (20.5, 25.4)	0.717
PaO_2_/FiO_2_, kPa	153.2 (127.6, 207.2)	123.1 (87.7, 172.9)	0.186
WBC (10^3^/uL)	8.2 (7.1, 10.7)	7.7 (5.5, 11.8)	0.307
Lymphocyte (10^3^/uL)	0.6 (0.5, 1.0)	0.8 (0.5, 0.9)	0.805
Platelet (10^3^/uL)	182.0 (134.3, 252.0)	164.0 (118.0, 208.0)	0.251
Albumin (g/dL)	3.8 (3.5, 4.1)	3.5 (3.2, 3.8)	0.016
Total bilirubin (mg/dL)	0.6 (0.4, 0.7)	0.8 (0.6, 1.2)	0.001
Bicarbonate (mmol/L)	22.0 (20.3, 24.0)	23.0 (21.0, 25.0)	0.382
Creatinine (mg/dL)	0.9 (0.7, 1.1)	0.8 (0.6, 1.0)	0.428
Troponin (pg/mL)	12.0 (9.3, 21.8)	12.5 (7.0, 31.3)	0.963
LDH (IU/L)	507.0 (396.3, 569.3)	527.0 (428.5, 674.5)	0.211
Procalcitonin (ng/mL)	0.4 (0.1, 0.7)	0.3 (0.1, 0.5)	0.782
Fibrinogen (mg/dL)	488.0 (414.8, 587.0)	445.0 (369.0, 615.0)	0.414
D-dimer (ng/mL)	487.0 (294.0, 651.5)	527.0 (294.0, 4301.5)	0.185
Ferritin (ng/mL)	455.6 (365.0, 985.8)	1098.2 (731.0, 1893.4)	0.003
CRP (mg/L)	111.6 (61.7, 144.6)	117.2 (59.7, 210.3)	0.490
IL-6 (pg/mL)	100.9 (18.9, 143.8)	65.1 (19.4, 191.5)	0.855
Soluble IL2 receptor (U/mL)	867.0 (726.8, 1073.8)	1036.0 (754.0, 1364.3)	0.135
Hospital day	15.0 (11.3, 19.0)	22.0 (18.0, 32.0)	<0.001
Mortality	3 (15.0)	7 (24.1)	0.496

Abbreviations: BMI, body mass index; WBC, white blood cell; LDH, lactate dehydrogenase; CRP, C-reactive protein; IL-6, interleukin-6; IL-2, interleukin-2; MV, mechanical ventilation.

**Table 4 jcm-10-04242-t004:** Cox regression analysis of risk factors related to in-hospital mortality in COVID-19 patients with respiratory failure.

	Univariable			Adjusted		
Variable	HR	95% CI	*p*-Value	HR	95% CI	*p*-Value
Age, years	1.077	0.983–1.180	0.113	1.023	0.942–1.112	0.585
Sex, male	1.622	0.405–6.493	0.495			
Initial P/F ratio	0.999	0.993–1.005	0.781	0.997	0.990–1.004	0.340
Δ WBC	1.102	0.970–1.251	0.136			
Δ lymphocyte	0.319	0.083–1.227	0.096	0.741	0.096–5.713	0.779
Δ Platelet	0.994	0.990–0.999	0.019	0.993	0.986–1.001	0.069
Δ LDH (IU/L)	1.003	0.999–1.006	0.140			
Δ Fibrinogen (mg/dL)	1.001	0.999–1.003	0.385			
Δ Ferritin (ng/mL)	0.999	0.997–1.001	0.502			
Δ CRP (mg/L)	1.003	0.996–1.011	0.367			
Δ IL-6/SD	1.664	0.896–3.088	0.107	1.234	0.594–2.564	0.573
Δ sIL-2R/SD	3.145	1.177–8.406	0.022	3.225	1.151–9.037	0.026

Abbreviations: WBC, white blood cell; LDH, lactate dehydrogenase; CRP, C-reactive protein; IL-6, interleukin-6; sIL-2R, soluble interleukin-2 receptor; SD, standard deviation. HR: hazard ratio; CI: confidence interval; Δ: Changes on Day 7 and Day 1

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
