# Peer review of "Soluble IL-2R Levels Predict in-Hospital Mortality in COVID-19 Patients with Respiratory Failure"

_jcm, 2021, doi:10.3390/jcm10184242_

Round 1

Reviewer 1 Report

 The paper is well designed and reported.

The originality is low, thus the authors should improve the points of originality of their work in the Discussion.

Author Response

Manuscript Title: Soluble IL-2R levels predict in-hospital mortality in COVID-19 patients with respiratory failure

We would like to thank all of the editors and reviewers for helping us make a better revision. We revised our manuscript according to the comments and recommendations of the reviewers. We highlighted all changes in the revised manuscript in blue letters. Here we include a separate itemized series of responses to the comments of the reviewers.

Responses to reviewers’ comments

# Reviewer 1

Comments to the Authors: The paper is well designed and reported.

The originality is low, thus the authors should improve the points of originality of their work in the Discussion.

Response : We appreciate your comments. We further described the strength of our study in discussion section. [page 12, line 208-213]

Reviewer 2 Report

Jang and colleagues studied a retrospective cohort of 49 patients with severe acute respiratory syndrome coronavirus 2. The main message is soluble IL-2R levels predict in-hospital mortality. This study is important. Although interesting, the manuscript has some limitations and there is place for minor improvements.

Soluble IL-2R assay method test is not described. Can you described it ?

L38: i don't understand mortality "between 65.7% and 94% in patients who receive mechanical ventilation". Wu and al. [2] demonstrate 94% of mortality. 

L39: "Recent studies" for Zhou et al. [3] and Henry [4] of 2020, March and June are maybe older.

L69: by the Society of Critical Care Medicine [29]. Citation n°29 is after n°8.

L86: Does the distribution of parameters is normal? 

Figure 3a and 3b: Can you add number of patient for each group?

L206: Hou et al. [14] use IL-2R/lymphocytes for predicting the clinical progression of patients with COVID-19. Can you more develop differences with your study?

L232: You already say this in L50. Gooding et al is cited twice [22] and [6].

L237: Zhang et al. is cited in double [26] and [27]

L242: "all registered patients were treated with the same protocol". Yes, it's very a good method in this study. 

Author Response

Manuscript Title: Soluble IL-2R levels predict in-hospital mortality in COVID-19 patients with respiratory failure

We would like to thank all of the editors and reviewers for helping us make a better revision. We revised our manuscript according to the comments and recommendations of the reviewers. We highlighted all changes in the revised manuscript in blue letters. Here we include a separate itemized series of responses to the comments of the reviewers.

# Reviewer 2

<Minor comments>

Comments to the Authors: Jang and colleagues studied a retrospective cohort of 49 patients with severe acute respiratory syndrome coronavirus 2. The main message is soluble IL-2R levels predict in-hospital mortality. This study is important. Although interesting, the manuscript has some limitations and there is place for minor improvements.

Comment 1: Soluble IL-2R assay method test is not described. Can you described it ?

Response 1: Thank you for your suggestion. Accordingly, we added sIL-2R assay method in method section as “The level of sIL-2R was measured using an automated immune chemiluminescent assay via IMMULITE ® 2000 XPi system (Siemens, USA).” [page 2, line 74-76]

Comment 2: L38: i don't understand mortality "between 65.7% and 94% in patients who receive mechanical ventilation". Wu and al. [2] demonstrate 94% of mortality.

Response 2: Thank you for your comment. We checked the paper again, and in clinical outcome section, they described 65.7% were dead who received mechanical ventilation. [page 1, line 38]

Comment 3: L39: "Recent studies" for Zhou et al. [3] and Henry [4] of 2020, March and June are maybe older.

Response 3: We appreciate your comments. We’ve changed “recent” to “certain” as your comment. [page 1, line 39]

 Comment 4: L69: by the Society of Critical Care Medicine [29]. Citation n°29 is after n°8.

Response 4: Thank you for your comment. We numbered all citations in order [page 2, line 72]

Comment 5: L86: Does the distribution of parameters is normal?

Response 5: Thank you for your comment. We changed the phrase as “according to the non-normal distribution of data”, [page 2, line 90-91]

Comment 6:  Figure 3a and 3b: Can you add number of patient for each group?

Response 6: Thank you for this kind comment. We add number of patients for each group in figure 3a and 3b. [page 11]

Comment 7:  L206: Hou et al. [14] use IL-2R/lymphocytes for predicting the clinical progression of patients with COVID-19. Can you more develop differences with your study?

Response 7: We appreciate this comment. We described the difference our study from the other study as “Hou et al[17]. found higher IL-2R-to-lymphocyte ratio to be related to the severity of COVID-19; however, the level of IL-2R itself was not statistically significant for the predic-tion of severe COVID‐19. Our study results confirmed this finding and showed that the level of sIL-2R is associated with survival as well as disease severity and that this was more applicable and easier to use in clinics, indicated by the PaO2/FiO2 ratio in patients with COVID-19.” [page 12, line 208-213]

Comment 8:   L232: You already say this in L50. Gooding et al is cited twice [22] and [6].

Response 8: Thank you for your comment. We deleted the sentence to avoid duplication. [page 12, line 234-236]

Comment 9:   L237: Zhang et al. is cited in double [26] and [27]

Response 9: Thank you for your comment. We deleted the doubled citation.

Comment 10:  L242: "all registered patients were treated with the same protocol". Yes, it's very a good method in this study. 

Response 10: Thank you for your comment.

Reviewer 3 Report

Main points:
• Introduction. In order to provide a complete background on the available inflammatory
biomarkers with prognostic significance in COVID-19, along with several cytokines and other
serum markers, novel cytometric biomarkers should also be mentioned, such as MDW and
NLR [Riva et al. Monocyte Distribution Width (MDW) as novel inflammatory marker with
prognostic significance in COVID-19 patients. Sci Rep. 2021.][Li et al. Predictive values of
neutrophil-to-lymphocyte ratio on disease severity and mortality in COVID-19 patients: a
systematic review and meta-analysis. Crit. Care 2020.].
• Lines 128-132: this different trend in sIL-2R levels between survivors and non-survivors
(basically, stable vs growing values, respectively) is an interesting observation, and a
statistical analysis could be applied to provide the p-value, possibly further supporting the
conclusions of this study.
• Lines 177-181: the Authors reported a ROC-derived threshold for sIL-2R, and thus could also
add the related AUC, sensitivity and specificity for this test.
• Figure 3a and 3b. In the legends, please specify whether the curves are calculated on IL-6/IL-
2R values at admission, at day +7, or both.
• Discussion. It would be interesting if the Authors could comment on the peculiar significance
of sIL-2R, possibly emerging as a novel serum biomarker of “T-cell immunosuppression” in
COVID-19 patients. Indeed, as discussed by the Authors, high levels of sIL-2R have been
linked to low activation state of lymphocytes, and this notion possibly finds further clinical
evidence in the lymphopenia typically observed in severe COVID-19 patients (here and
previously reported with elevated sIL-2R levels). As a matter of fact, severe COVID-19
should be seen as a complex virus-associated disease, characterized not only by a lifethreatening
‘cytokine storm’, but also by a relevant impairment of protective T-cell functions
[Riva et al. COVID-19: more than a cytokine storm. Crit. Care 2020.][Remy et al.
Immunotherapies for COVID-19: lessons learned from sepsis. Lancet Respir. Med. 2020.].
To date, most of the serum biomarkers used as prognostic factors in COVID-19 are
‘activation’ markers (such as IL-6, CRP, PCT, etc), which provide valuable information on
the hyper-inflammatory state of severe COVID-19 (i.e., cytokine storm). On the other side,
routine laboratory tests investigating T-cell immunosuppressive state of COVID-19 patients
are needed: assessment of T-cell exhaustion markers by flow cytometry is not yet applicable
in the clinical practice, and basically, only total lymphocyte count is currently available to this
aim. In this view, sIL-2R may represent a new valuable tool to monitor the
immunosuppression level in COVID-19, and could usefully be combined with classic markers
of inflammation.
Minor corrections:
- lines 9-10: S.H.L. is duplicated.
- lines 17-18: the abstract reports that the study is based on 43 patients enrolled from Feb 2020 to Jan
2021, while in the text 49 patients are described (line 103 and in all tables/figures), as well as the
enrollment period becomes Feb 2020 – Apr 2021.
- lines 25-26: the sentence is not clear, please rephrase or delete “, not mechanical ventilator use,”.
- lines 142 and 143: “IL-2Rc” is not defined.
- reference 29 seems to be not cited in the text

Author Response

Manuscript Title: Soluble IL-2R levels predict in-hospital mortality in COVID-19 patients with respiratory failure

We would like to thank all of the editors and reviewers for helping us make a better revision. We revised our manuscript according to the comments and recommendations of the reviewers. We highlighted all changes in the revised manuscript in blue letters. Here we include a separate itemized series of responses to the comments of the reviewers.

# Reviewer 3

Comment 1: Introduction. In order to provide a complete background on the available inflammatory biomarkers with prognostic significance in COVID-19, along with several cytokines and other serum markers, novel cytometric biomarkers should also be mentioned, such as MDW and NLR [Riva et al. Monocyte Distribution Width (MDW) as novel inflammatory marker with prognostic significance in COVID-19 patients. Sci Rep. 2021.][Li et al. Predictive values of neutrophil-to-lymphocyte ratio on disease severity and mortality in COVID-19 patients: a systematic review and meta-analysis. Crit. Care 2020.].

Response 1: Thank you for your suggestion. Accordingly, we added further information in the introduction section as “New cytometric parameters, such as Monocyte Distribution Width (MDW) and Neutrophil-to-Lymphocyte Ratio (NLR), were recently found to correlate well with the clinical severity of COVID-19-associated inflammatory disorder.” [page 2, line 46-48]

Comment 2: Lines 128-132: this different trend in sIL-2R levels between survivors and non-survivors (basically, stable vs growing values, respectively) is an interesting observation, and a statistical analysis could be applied to provide the p-value, possibly further supporting the conclusions of this study.

Response 2: Thank you for your comment. We added the p-value in the result section. [page 5, line 135-138]

Comment 3:  Lines 177-181: the Authors reported a ROC-derived threshold for sIL-2R, and thus could also add the related AUC, sensitivity and specificity for this test.

Response 3: We appreciate your comments. We further reported the AUC area, sensitivity, and specificity value. [page 10, line 182-185]

Comment 4: Figure 3a and 3b. In the legends, please specify whether the curves are calculated on IL-6/IL-2R values at admission, at day +7, or both.

Response 4: Thank you for your comments. We described the day of the laboratory test in the figure legends. [page 10, line 189 and 191]

Comment 5: Discussion. It would be interesting if the Authors could comment on the peculiar significance of sIL-2R, possibly emerging as a novel serum biomarker of “T-cell immunosuppression” in COVID-19 patients. Indeed, as discussed by the Authors, high levels of sIL-2R have been linked to low activation state of lymphocytes, and this notion possibly finds further clinical evidence in the lymphopenia typically observed in severe COVID-19 patients (here and previously reported with elevated sIL-2R levels). As a matter of fact, severe COVID-19 should be seen as a complex virus-associated disease, characterized not only by a lifethreatening cytokine storm’, but also by a relevant impairment of protective T-cell functions [Riva et al. COVID-19: more than a cytokine storm. Crit. Care 2020.][Remy et al.

Immunotherapies for COVID-19: lessons learned from sepsis. Lancet Respir. Med. 2020.].

To date, most of the serum biomarkers used as prognostic factors in COVID-19 are activation’ markers (such as IL-6, CRP, PCT, etc), which provide valuable information on the hyper-inflammatory state of severe COVID-19 (i.e., cytokine storm). On the other side, routine laboratory tests investigating T-cell immunosuppressive state of COVID-19 patients are needed: assessment of T-cell exhaustion markers by flow cytometry is not yet applicable in the clinical practice, and basically, only total lymphocyte count is currently available to this aim. In this view, sIL-2R may represent a new valuable tool to monitor the immunosuppression level in COVID-19, and could usefully be combined with classic markers of inflammation.

Response 5: Thank you for your comment. Accordingly, we further described the detailed contents in the discussion section as “SARS-CoV-2 has been known to lead to hyperinflammation as well as T-cell deficiencies, associated with life-threatening organ dysfunction”. [page 12, line 232-234]

Minor corrections

Comment 1:  lines 9-10: S.H.L. is duplicated.

Response 1: We appreciate this comment. Sang Hoon Lee (S.H.L, [email protected]) and Su Hwan Lee (S.H.L, [email protected]) were both authors of this study.

Comment 2:  lines 17-18: the abstract reports that the study is based on 43 patients enrolled from Feb 2020 to Jan 2021, while in the text 49 patients are described (line 103 and in all tables/figures), as well as the enrollment period becomes Feb 2020 – Apr 2021.

Response 2: We appreciate this comment. We changed the number of recruited patients as “forty-nine” in the abstract and the period as “Feb 2020 to Apr 2021”. [page 1, line 17-18]

Comment 3:   - lines 25-26: the sentence is not clear, please rephrase or delete “, not mechanical ventilator use,”.

Response 3: Thank you for your comment. We deleted it to avoid misunderstanding. [page 1, line 26]

Comment 4:   lines 142 and 143: “IL-2Rc” is not defined.

Response 4: Thank you for your comment. We were meant to describe sIL-2R. Accordingly, we changed sIL-2Rc to sIL-2R. [page 6, line 144-147]

Comment 5:  - reference 29 seems to be not cited in the text

Response 5: We appreciate this comment. The reference numbered [31] were cited in the discussion section. [page 12, line 242]